# Preoperative antihypertensives and hypotension during bladder tumor resection with oral 5-aminolevulinic acid administration

Yuriko Kondo[1,2], Takahiro Mihara[1]*, Nanako Yoshikawa[2], Noriyuki Echigo[2], Yusuke Nagamine[3], Takahisa Goto[3]

**1** Graduate School of Data Science, Department of Health Data Science, Yokohama City University, Yokohama, Kanagawa, Japan, **2** Department of Anesthesiology, Yokohama Rosai Hospital, Yokohama, Kanagawa, Japan, **3** Anesthesiology and Critical Care Medicine, Yokohama City University School of Medicine Graduate School of Medicine, Yokohama, Kanagawa, Japan

* meta.analysis.r@gmail.com

## Abstract

5-Aminolevulinic acid hydrochloride (5-ALA), a photodynamic diagnostic agent, visualizes bladder cancer. Previous research has indicated that preoperative intake of 5-ALA leads to a higher incidence of hypotension. Particularly in patients with hypertension, suggestions include discontinuing antihypertensive medications on the morning of surgery to prevent hypotension. However, the effects of antihypertensive drugs on hypotension in patients taking 5-ALA before surgery remains unexamined. We conducted a single-center observational study that included patients aged 20 and above who were regularly taking antihypertensives and underwent transurethral resection of bladder tumors (TURBT) after taking 5-ALA. Patients who took antihypertensives on the morning of surgery were defined as the continued group, whereas those who did not were defined as the discontinued group. Hypotension was defined as a mean blood pressure (MBP) of less than 65 mmHg for 20 min or longer. To adjust for confounding factors, we used propensity scores for inverse probability weighting and performed modified Poisson regression analysis to calculate risk ratios (RRs) and 95% confidence intervals (CIs). We analyzed 132 cases. The crude incidence of hypotension was higher in the continued group compared to the discontinued group (33/51 [64.7%] vs 38/81 [46.9%]; RR 1.38, 95% CI 1.01–1.88; p = 0.041). However, no significant difference was observed between groups after adjustment (RR 1.05, 95% CI 0.66–1.68). In conclusion, the adjusted results suggested no significant association between the continuation of antihypertensive medication and the incidence of intraoperative hypotension. No substantial justification was provided for routinely discontinuing antihypertensive medications.

## Introduction

5-Aminolevulinic acid hydrochloride (5-ALA), a photodynamic diagnostic agent, visualizes bladder cancer. Administered orally 2–3 h before transurethral resection of bladder tumor (TURBT), 5-ALA is expected to visualize bladder tumors and reduce residual tumors [1]. One

**Data availability statement:** There are ethical restrictions which prevent the public sharing of minimal data for this study because the data contain sensitive participant information. Data are available upon request from Yokohama Rosai Hospital's Research Ethics Committee via email (rinri@yokohamah.johas.go.jp) for researchers who meet the criteria for access to confidential data.

**Funding:** The author(s) received no specific funding for this work.

**Competing interests:** The authors have declared that no competing interests exist.

of the side effects of 5-ALA is hypotension. Several case reports document severe hypotension in the perioperative period after taking 5-ALA [2–4]. Prior studies have indicated that the intake of 5-ALA before surgery leads to a higher incidence of hypotension [5–8] and increases the frequency or dosage of intraoperative vasoactive agents used [7]. The frequency of hypotension in patients taking 5-ALA varies in studies, ranging from 6–70% [9–11].

Several studies have discussed risk factors for hypotension in patients receiving 5-ALA. General anesthesia, age, body mass index (BMI), renal dysfunction, regular use of antihypertensive medications, sex, and cardiovascular disease have been identified as potential risk factors [8–13]. However, several potential risk factors, apart from general anesthesia, are patient characteristics challenging to modify preoperatively. Further exploration of modifiable preoperative risk factors would aid the discourse on preventing hypotension after 5-ALA administration.

A previous study investigated patients who preoperatively took 5-ALA for upper urinary tract tumor surgery and observed that a significant number of those who experienced hypotension were taking antihypertensive drugs preoperatively [14]. Another study concluded that discontinuation of antihypertensives should be considered to prevent hypotension [15]. However, these studies did not primarily focus on the association between continuing antihypertensive medication and hypotension and did not adjust for confounding factors. Therefore, assessing the association between continuing antihypertensive medication and hypotension requires appropriate adjustments for confounding factors.

This cohort study aimed to evaluate the effect of continuing morning antihypertensives on intraoperative hypotension in patients who were regularly taking antihypertensives and undergoing TURBT following 5-ALA administration.

## Materials and methods

### Study design and setting

This single-center observational study was conducted at the Yokohama Rosai Hospital, a general hospital in Japan. This study was approved by the Ethics Committee of Yokohama Rosai Hospital (approval number 2–15 [Chairperson Dr. Youji Mikami, 2 July 2020], Yokohama, Kanagawa, Japan). For eligible patients who underwent surgery between October 2018 and July 2020, opt-outs were made on the institution's website. Patients undergoing surgery after July 2020 were informed about the study during their anesthesiology consultation, and written consent was obtained. The data were accessed for research purposes most recently on December 27, 2022. The authors retained access to information that could identify individual participants after data collection. All methods were performed in accordance with the Declaration of Helsinki. The manuscript was prepared following the recommendations of the Strengthening the Reporting of Observational Studies in Epidemiology (STROBE) statement [16].

Although we routinely administered antihypertensive medications to patients undergoing TURBT after 5-ALA on the morning of surgery, mirroring our approach for other surgical cases, due to concerns about perioperative hypotension, antihypertensive medications were discontinued on the day of surgery in March 2020 following discussions between the director of the department of anesthesiology and urologists.

### Patient selection

We included patients aged 20 years or older undergoing scheduled TURBT after 5-ALA at Yokohama Rosai Hospital between October 2018 and March 2022. Our analysis focused on patients who regularly used antihypertensive medications such as calcium channel blockers, beta-blockers, angiotensin-converting enzyme inhibitors (ACE-I), angiotensin receptor

blockers (ARBs), alpha-blockers, and diuretics. We excluded cases whose operative duration was less than 20 min. This exclusion criterion was based on our study's definition of hypotension, which was a mean blood pressure (MBP) of less than 65 mmHg for ≥20 min. Additionally, we excluded cases taking ACE-Is, ARBs, or compounding agents containing ACE-Is or ARBs as single agents. This exclusion was based on the prevalent practice among our hospital's anesthesiologists, who were not administering these drugs on the morning of surgery.

All patients received approximately 20 mg/kg 5-ALA orally 2–3 h before surgery. The patients were instructed to fast from the evening before surgery and to abstain from drinking water 2–3 h before surgery. The method of anesthesia induction for each patient was determined by the anesthesiologist in charge. Vasoactive drugs were administered postoperatively at the discretion of an intensive care physician.

## Exposure and outcomes

The exposure was antihypertensives on the morning of surgery. Patients taking antihypertensives were labeled the continued group, whereas those not taking them were the discontinued group. In cases where a patient regularly took more than one type of antihypertensive medication, they were classified in the continued group if at least one medication was taken on the morning of surgery.

The primary outcome measure was the incidence of intraoperative hypotension. Hypotension was defined as MBP of less than 65 mmHg that persisted for ≥20 min. The definition of hypotension was preliminarily determined based on previous studies that intraoperative MBP below 60–70 mmHg is associated with myocardial infarction, acute kidney injury (AKI), and death [17] and that intraoperative MBP below 65 mmHg for 20 min or longer increases the risk of AKI and myocardial infarction [18]. Intraoperative vital signs were extracted as electronic data from the automated electronic anesthesia record. Secondary outcomes were the postoperative hypotension, presence of postoperative AKI, and adverse events during hospitalization (syncope, dizziness, cold sweats, preoperative nausea and vomiting, postoperative nausea and vomiting, intensive care unit (ICU) admission, new angina, new arrhythmia, postoperative abnormal hypertension, disturbance of consciousness, cerebral infarction/bleeding, myocardial infarction, heart failure, cardiac death, and death). Based on a previous study, postoperative hypotension were defined as systolic blood pressure (SBP) less than 90 mmHg [19]. The postoperative period was defined as the period from leaving the operating room to the day after surgery. Postoperative AKI was defined as an increase of ≥0.3 mg/dl creatinine (Cr) within 48 h or an increase of ≥1.5 times from the baseline Cr value within 7 days, adopting the Cr values of the Kidney Disease Improving Global Outcomes (KDIGO) criteria [20]. Postoperative abnormal hypertension was defined as SBP of 180 mmHg or higher or diastolic blood pressure of 110 mmHg or higher, which is defined as degree III hypertension by the Japanese guidelines [21]. Postoperative vital signs (non-invasive blood pressure and heart rate) and physical symptoms were extracted from the electronic medical record. The timing of the vital signs survey was as follows: the day before surgery; before taking 5-ALA on the day of surgery; before leaving the ward; upon returning to the ward; 3, 6, and 12 h after returning to the ward; and the day after surgery.

## Data extraction

The type of anesthesia, duration of anesthesia, duration of surgery, presence or absence and amount of intraoperative and postoperative vasoactive agents (ephedrine, phenylephrine, dopamine, dobutamine, noradrenaline, adrenaline, and vasopressin) used, presence or absence and amount of atropine used, intraoperative infusion volume, and 24-h infusion

volume were extracted from the electronic anesthesia and medical records. We manually extracted data from electronic medical records to determine whether the patients took antihypertensives. Data extraction items were as follows: age, sex, height, weight, BMI, American Society of Anesthesiologists physical status (ASA-PS), comorbidities (hypertension, myocardial infarction, heart failure, arrhythmia, valvular disease, vascular disease, chronic respiratory disease, chronic renal dysfunction, dialysis, diabetes, liver dysfunction, cerebrovascular disease, collagen disease, depression, schizophrenia), Charlson comorbidity index (CCI), smoking history, preoperative blood test data (albumin (Alb), C-reactive protein (CRP), hemoglobin (Hb), hematocrit (Ht), blood urea nitrogen (BUN), Cr, estimated glomerular filtration rate (eGFR), type of regular oral medication, 5-ALA dosage, and date of surgery. Data were manually extracted from electronic medical records. Chronic renal dysfunction was defined as an eGFR less than 60 calculated from the preoperative Cr level, based on the literature [22].

## Statistical analysis

Descriptive statistics were computed for each variable and incidence of hypotension. Categorical variables were presented as numbers and percentages, whereas continuous variables were depicted with means and standard deviations or medians and interquartile ranges. Univariate analysis of comparisons between the two groups was performed for each variable. $\chi^2$ tests were performed for categorical variables, and t-tests or Wilcoxon rank sum tests for continuous variables. The normality of distribution was visually determined using a histogram. Univariate modified Poisson regression analysis was performed, and the risk ratio (RR) and 95% confidence interval (CI) for intraoperative hypotension were calculated.

Confounding factors were adjusted using the propensity score (PS) method. A logistic regression model was used to estimate the propensity scores. The covariates for the PS model were age, sex, BMI, time of surgery (year and month of surgery as continuous variables), diabetes, vascular disease, heart failure, hypertension, smoking status, eGFR, arrhythmia, stroke, ASA classification, and general anesthesia. The covariates were selected based on their relevance to previous studies and clinical knowledge. Weighting was performed to estimate the average treatment effect on treated (ATT). Specifically, we assigned a weight of "1" to the continued group and a "PS/(1-PS)" to the discontinued group. The discriminability of the groups by PS was confirmed using C-statistics. We described the standardized mean difference (SMD) between the groups of covariates after PS weighting. We defined no difference between the groups as an SMD < 0.1. Generalized estimating equations were used to calculate the RR and 95% CIs for intraoperative hypotension for antihypertensive medication on the morning of surgery [23,24].

Additionally, we used a weighting method different from that used in the main analysis to calculate the RR and 95% CI. This alternative weighting aimed at estimating the average treatment effect for the overlap population (ATO). In the overlap weighting method, treated patients are weighted by the probability of not receiving treatment (1 − PS), and untreated patients are weighted by the probability of receiving the treatment (PS). These weights are smaller for extreme PS values, such that outliers that are nearly always treated (PS near 1) or never treated (PS near 0) do not dominate the results or worsen precision, as occurs with inverse probability weighting. Overlap weighting overcomes the limitations of conventional inverse probability weighting, which excessively weighs patients whose PS is close to 1 when they are assigned to the control group [25].

The secondary outcomes, incidence of postoperative hypotension, presence of postoperative AKI, and adverse events during hospitalization were described for each group. No measurements were missing for any variables. Statistical significance was set at P < 0.05. The

R statistical software package version 4.2.0 (R Foundation for Statistical Computing, Vienna, Austria) was used for the statistical analysis.

## Results

A total of 132 cases (108 patients) met the inclusion criteria. All patients were included in the analysis (Fig 1).

The discontinued and continued groups included 81 and 51 cases, respectively. The background of each group is presented in Table 1. Patients in the discontinued group were slightly younger and had more cases of ASA-PS 3. Regarding comorbidities, diabetes mellitus was more frequent in the discontinued group, and arrhythmia was more frequent in the continued group. Beta-blockers were more frequently used in the continued group. The surgical time, anesthesia time, and method of anesthesia were not significantly different between the two groups (Table 1). In the discontinued group, 91.4% of surgeries were performed after March 2020, whereas 17.6% of surgeries were performed after March 2020 in the continued group. The medical records of seven patients in the discontinued group who underwent surgery before March 2020 and nine patients in the continued group after March 2020 were reviewed;

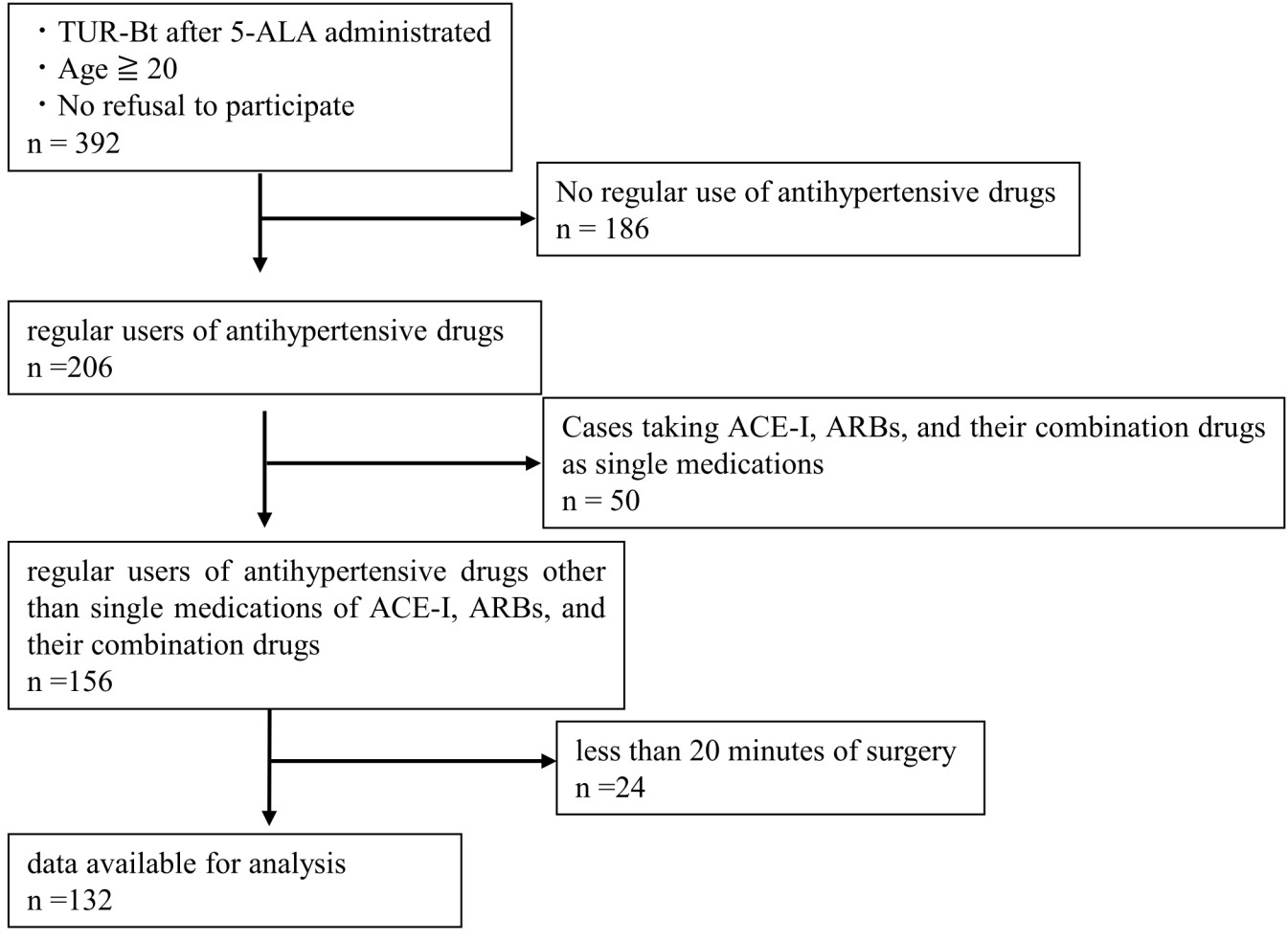

**Fig 1. Diagram of patient selection.**

**Table 1. Patient characteristics.**

| | Discontinued group (81) | Continued group (51) | SMD | p value | after adjustment | | |
|---|---|---|---|---|---|---|---|
| | | | | | Discontinued group | Continued group | SMD |
| Age, years | 74.9 ± 8.0 | 77.3 ± 6.1 | 0.329 | 0.076 | 77.1 ± 7.4 | 77.3 ± 6.1 | 0.021 |
| Male, n (%) | 69 (85.2) | 42 (82.4) | 0.028 | 0.850 | 32.6 (72.4) | 42.0 (82.4) | 0.099 |
| BMI, kg/m$^2$ | 24.4 (22.5–27.0) | 23.2 (21.2–25.2) | 0.376 | 0.050 | 23.7 (22.5–24.0) | 23.2 (20.9–25.2) | 0.090 |
| 5-ALA dose, mg/kg | 19.8 (19.5–20.0) | 19.9 (19.6–20.0) | 0.254 | 0.263 | | | |
| ASA-PS, n (%) | | | | 0.321 | | | |
| 1 | 1 (1.2) | 0 (0.0) | 0.012 | | 0.0 (0.0) | 0.0 (0.0) | 0.0 |
| 2 | 72 (88.9) | 49 (96.1) | 0.072 | | 44.6 (99.3) | 49.0 (96.1) | 0.032 |
| 3 | 8 (9.9) | 2 (3.9) | 0.060 | | 0.3 (0.7) | 2.0 (3.9) | 0.032 |
| Myocardial infarction, n (%) | 12 (14.8) | 7 (13.7) | 0.011 | >0.999 | | | |
| Heart failure, n (%) | 1 (1.2) | 1 (2.0) | 0.007 | >0.999 | 0.6 (1.3) | 1.0 (2.0) | 0.007 |
| Vascular disease, n (%) | 10 (12.3) | 9 (17.6) | 0.053 | 0.555 | 4.9 (11.0) | 9.0 (17.6) | 0.067 |
| Diabetes mellitus, n (%) | 22 (27.2) | 7 (13.7) | 0.134 | 0.110 | 2.6 (5.9) | 7.0 (13.7) | 0.079 |
| Impairment of liver function, n (%) | 12 (14.8) | 8 (15.7) | 0.009 | >0.999 | | | |
| Chronic renal dysfunction, n (%) | 42 (51.9) | 31 (60.8) | 0.089 | 0.409 | | | |
| Hemodialysis | 1 (1.2) | 0 (0.0) | 0.012 | >0.999 | | | |
| Cerebral stroke, n (%) | 12 (14.8) | 11 (21.6) | 0.068 | 0.447 | 4.5 (10.0) | 11.0 (21.6) | 0.116 |
| Smoking, n (%) | 16 (19.8) | 12 (23.5) | 0.038 | 0.766 | 6.7 (14.9) | 12.0 (23.5) | 0.087 |
| Hypertension, n (%) | 79 (97.5) | 50 (98.0) | 0.005 | >0.999 | 44.2 (98.3) | 50.0 (98.0) | 0.003 |
| Arrhythmia, n (%) | 12 (14.8) | 16 (31.4) | 0.166 | 0.041 | 12.3 (27.3) | 16.0 (31.4) | 0.041 |
| Valvular disease, n (%) | 2 (2.5) | 2 (3.9) | 0.015 | >0.999 | | | |
| CCI | 3.0 (2.0–4.0) | 3.0 (2.0–4.0) | 0.073 | 0.885 | | | |
| eGFR, mL/min/1.73 m$^2$ | 57.1 ± 17.4 | 57.3 ± 16.2 | 0.008 | 0.964 | 61.1 ± 12.8 | 57.3 ± 16.2 | 0.236 |
| Mean blood pressure on the day before surgery, mmHg | 93.0 ± 11.7 | 91.8 ± 10.7 | 0.106 | 0.558 | | | |
| Types of regular use antihypertensives | | | | | | | |
| Calcium channel blockers, n (%) | 73 (90.1) | 46 (90.2) | 0.001 | >0.999 | | | |
| Alpha-blockers, n (%) | 5 (6.2) | 3 (5.9) | 0.003 | >0.999 | | | |
| Beta-blockers, n (%) | 15 (18.5) | 15 (29.4) | 0.109 | 0.215 | | | |
| Diuretics, n (%) | 3 (3.7) | 2 (3.9) | 0.002 | >0.999 | | | |
| ACE-I, ARB, n (%) | 31 (38.3) | 20 (39.2) | 0.009 | >0.999 | | | |
| Compounding agents, n (%) | 5 (6.2) | 2 (3.9) | 0.023 | 0.870 | | | |
| Number of regularly used antihypertensives, n (%) | 5 (6.2) | | | 0.832 | | | |
| 1 | 38 (46.9) | 22 (43.1) | 0.038 | | | | |
| 2 | 36 (44.4) | 22 (43.1) | 0.013 | | | | |
| 3 | 6 (7.4) | 6 (11.8) | 0.044 | | | | |
| 4 | 1 (1.2) | 1 (2.0) | 0.007 | | | | |
| Types of antihypertensives on the morning of surgery | | | | | | | |
| Calcium channel blockers, n (%) | | 42 (82.4) | | | | | |
| Alpha-blockers, n (%) | | 0 (0.0) | | | | | |
| Beta-blockers, n (%) | | 13 (25.5) | | | | | |
| Diuretics, n (%) | | 2 (3.9) | | | | | |
| ACE-I, ARB, n (%) | | 3 (5.9) | | | | | |

*(Continued)*

**Table 1.** (Continued)

| | Discontinued group (81) | Continued group (51) | SMD | p value | after adjustment | | |
| --- | --- | --- | --- | --- | --- | --- | --- |
| | | | | | Discontinued group | Continued group | SMD |
| Compounding agents, n (%) | | 0 (0.0) | | | | | |
| Details of surgery and anesthesia | | | | | | | |
| Operative time, min | 46.0 (29.0–64.0) | 41.0 (33.0–60.5) | 0.048 | 0.905 | | | |
| Anesthesia time, min | 82.0 (62.0–98.0) | 81.0 (64.5–93.5) | 0.059 | 0.987 | | | |
| General anesthesia, n (%) | 80 (98.8) | 50 (98.0) | 0.007 | >0.999 | 44.9 (99.8) | 50.0 (98.0) | 0.018 |
| month of surgery§ | 28.0 (20.0–35.0) | 11.0 (5.0–15.0) | 1.630 | <0.001 | 9.8 (1.7–19.3) | 11.0 (5.0–15.0) | 0.016 |
| Cases with surgery dates after March 2020, n (%) | 74 (91.4) | 9 (17.6) | 0.737 | <0.001 | | | |

*Number (%), Median (interquartile range), Mean ± standard deviation.

SMD, standardized mean difference; BMI, body mass index; 5-ALA, 5-aminolevulinic acid; ASA-PS, American Society of Anesthesiologists physical status; CCI, Charlson comorbidity index; ACE-I, angiotensin-converting enzyme inhibitor, ARB: angiotensin II receptor blocker.

§A continuous variable assigned monthly, with October 2018 designated as 1.

however, the reasons were not noted. The interval between blood pressure measurements was typically 2.5–5 min.

The discontinued group had a crude hypotension incidence of 38/81 (46.9%), whereas the continued group had 33/51 (64.7%), indicating a significantly higher occurrence (RR 1.38; 95% CI 1.01–1.88; p = 0.041; Table 2). After weighting, all covariates had SMD < 0.1 except for eGFR (61.1 ± 12.8 vs 57.3 ± 16.2, SMD = 0.236) and stroke (10.0% vs 21.6%, SMD = 0.116, Fig 2). The C statistic for this PS model was 0.899 (95% CI 0.843–0.956). After adjusting, the RR was 1.05 (95% CI 0.66–1.68, p = 0.843, Table 2). The overlap weighting method showed that the two groups were well balanced (SMD < 0.1 for all variables, S1 Fig); the results were similar to those in the main analysis. (S1 Table).

The intraoperative use of vasoactive agents is shown in Table 3, with the continued group showing a higher rate of intraoperative atropine use after adjustment. For secondary outcomes, after adjustment, no statistically significant differences existed for each item between the groups (Table 4).

The intraoperative mean blood pressure is shown in Fig 3.

## Discussion

In this study, no significant association was found between continuing or discontinuing antihypertensive medications on the morning of surgery and intraoperative hypotension during scheduled TURBT following 5-ALA administration. Additionally, the incidence of adverse events during hospitalization did not differ significantly between the two groups. These findings suggest that there is no substantial justification for routinely discontinuing antihypertensive medication on the morning of surgery.

One possible explanation for this result is the residual effect of long-acting antihypertensive drugs. Regular intake of antihypertensives with long-half-lives, even when discontinued on the morning of surgery, may result in persistent pharmacologic effects such as vasodilation, which can contribute to prolonged antihypertensive effects during the perioperative period. Previous studies have identified long-acting ACE-Is and ARBs as risk factors for hypotension, even after a 24-hour withdrawal period [26,27]. For example, severe hypotension has been reported in patients taking amlodipine, a calcium channel blocker with a half-life of approximately 40 hours, despite discontinuation on the morning of surgery [27]. In our study, more

**Table 2. Incidence of intraoperative hypotension.**

| | Discontinued group (%) | Continued group (%) | Risk ratio (95% confidence interval) | p value |
|---|---|---|---|---|
| Crude model | 46.9 | 64.7 | 1.38 (1.01–1.88) | 0.041 |
| Adjusted model | 61.7 | 64.7 | 1.05 (0.66–1.68) | 0.843 |

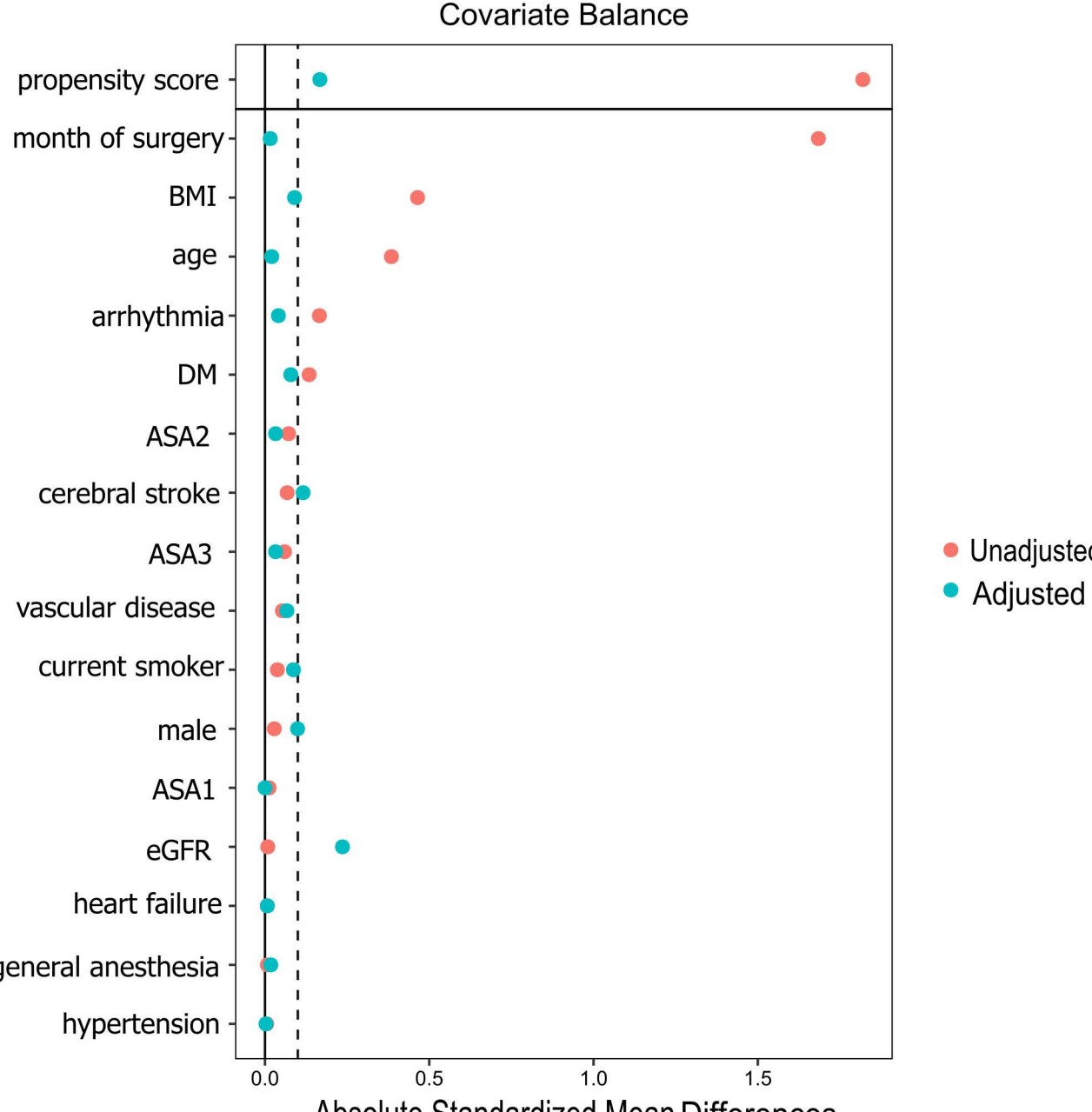

**Fig 2. Standardized mean difference of variables before and after propensity score weighting aimed at estimating the average treatment effect on the treated.** Red dots represented crude standardized mean difference, and blue dots represented standardized mean difference after weighting.

**Table 3. Use of intraoperative vasoactive agents.**

| | Crude | | | | Adjusted model (ATT) | | | |
|---|---|---|---|---|---|---|---|---|
| | Discontinued group (%) | Continued group (%) | RR (95% CI) | p value | Discontinued group (%) | Continued group (%) | RR (95% CI) | p value |
| Ephedrine (%) | 67.9 | 58.8 | 0.87 (0.66–1.14) | 0.305 | 61.0 | 58.8 | 0.97 (0.55–1.70) | 0.901 |
| Phenylephrine (%) | 71.6 | 58.8 | 0.82 (0.63–1.07) | 0.150 | 70.7 | 58.8 | 0.83 (0.55–1.27) | 0.390 |
| Atropine (%) | 17.3 | 21.6 | 1.25 (0.62–2.53) | 0.540 | 4.1 | 21.6 | 5.25 (1.61–17.1) | 0.006 |
| Noradrenaline (%) | 3.7 | 0.0 | – | – | 0.1 | 0.0 | – | – |
| Adrenaline (%) | 1.2 | 0.0 | – | – | 0.0 | 0.0 | – | – |

Dopamine, dobutamine, and vasopressin were not used. CI, confidence interval.

**Table 4. Adverse events during hospitalization.**

| | Crude | | | | Adjusted model (ATT) | | | |
|---|---|---|---|---|---|---|---|---|
| | Discontinued group (%) | Continued group (%) | RR (95% CI) | p value | Discontinued group (%) | Continued group (%) | RR (95% CI) | p value |
| Postoperative hypotension (%) | 4.9 | 5.9 | 1.19 (0.28–5.11) | 0.814 | 2.0 | 5.9 | 2.92 (0.44–19.5) | 0.268 |
| AKI (%) | 0.0 | 2.0 | – | – | 0.0 | 2.0 | – | – |
| Dizzy (%) | 1.2 | 0.0 | – | – | 0.2 | 0.0 | – | – |
| Cold sweats (%) | 1.2 | 0.0 | – | – | 0.2 | 0.0 | – | – |
| Preoperative nausea and vomiting (%) | 6.2 | 3.9 | 0.64 (0.13–3.15) | 0.579 | 1.6 | 3.9 | 2.44 (0.38–15.6) | 0.346 |
| Postoperative nausea and vomiting (%) | 30.9 | 41.2 | 1.33 (0.84–2.12) | 0.222 | 46.3 | 41.2 | 0.89 (0.43–1.85) | 0.755 |
| Admission to the ICU (%) | 2.5 | 0.0 | – | – | 1.5 | 0.0 | – | – |
| Arrhythmia (%) | 2.5 | 0.0 | – | – | 1.7 | 0.0 | – | – |
| Postoperative hypertension (%) | 3.7 | 3.9 | 1.06 (0.18–6.12) | 0.949 | 1.1 | 3.9 | 3.54 (0.50–25.2) | 0.207 |

Disturbances in consciousness, cerebral infarction, cerebral hemorrhage, myocardial infarction, heart failure, or death were not observed. CI, confidence interval.

than half of the patients were on amlodipine or combination drugs containing amlodipine, while a smaller proportion were administered telmisartan, an ARB with a long half-life. In patients regularly using long-acting antihypertensives, a withdrawal periods of 12–24 h may be insufficient to eliminate their effects on intraoperative blood pressure. In our study, withdrawal periods varied between patients, with some experiencing approximately 28 hours and others about 12 hours, depending on their usual medication schedule. This study evaluated the approach of uniformly discontinuing antihypertensive medications on the day of surgery, a method chosen for its high clinical feasibility and simplicity. However, this approach did not effectively prevent hypotension, highlighting the need for considering withdrawal periods based on the half-life of medications. Future research should evaluate the impact of drug type, half-life, and timing of withdrawal on the risk of hypotension.

Another potential factor is the potent hypotensive effect of 5-ALA itself, which when compounded by anesthetic drugs, may outweigh the influence of preoperative antihypertensive medication withdrawal. While our study was not specifically designed to evaluate this interaction, the current findings highlight the need for tailored perioperative management strategies, such as adjusting the timing of antihypertensive withdrawal or implementing pharmacologic interventions.

The pathogenesis of hypotension following 5-ALA administration remains unclear, but animal studies suggest that its metabolite, that protoporphyrin IX (Pp IX), activates soluble

(a)

(b)

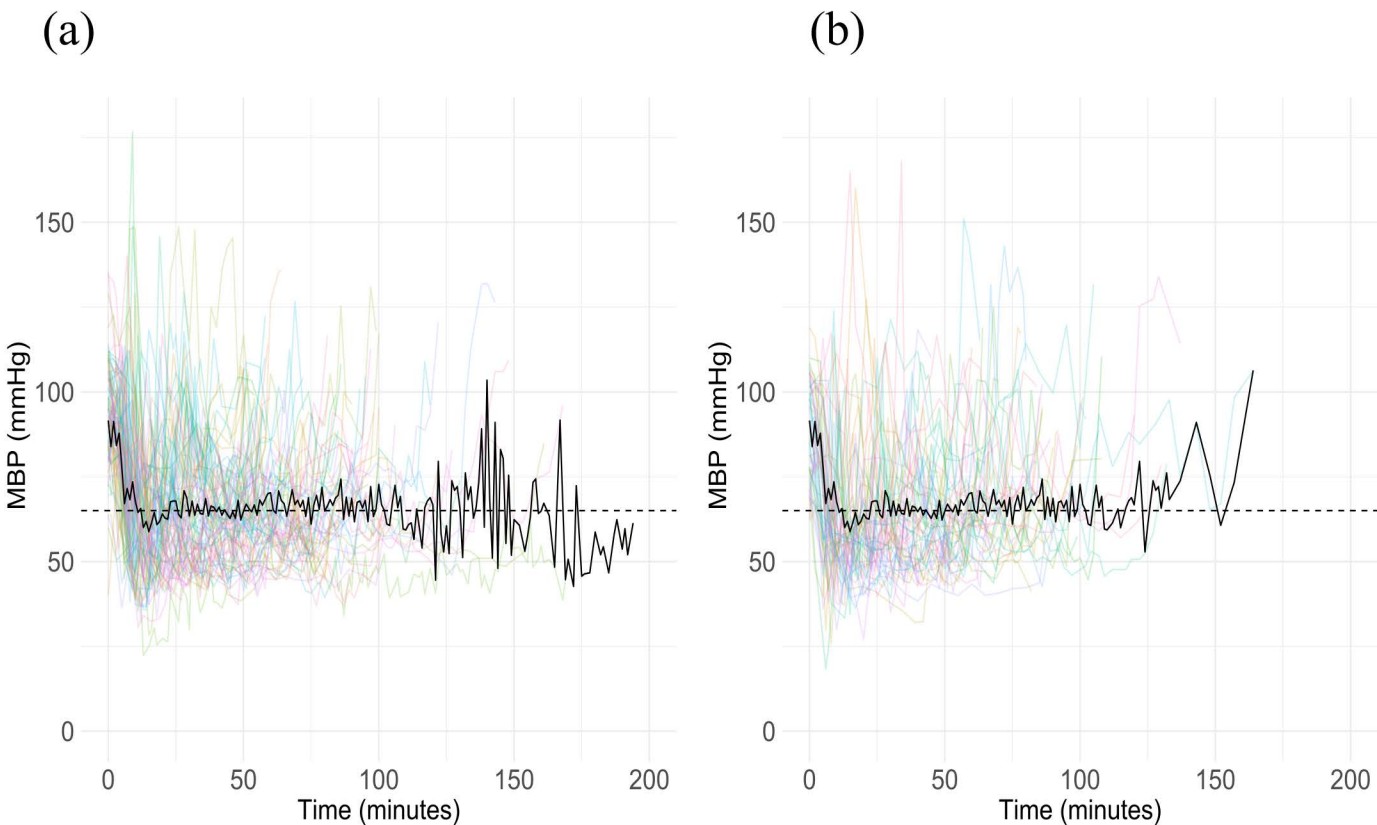

**Fig 3. Mean blood pressure in each case.** (a) discontinued group (b) continued group, MBP; mean blood pressure, Dashed line; 65mmHg, Black line; mean of MBP, Colored lines; MBP of individual cases.

guanylate cyclase, leading to vasodilation of systemic and pulmonary vessels [28]. Additionally, increased vascular permeability after 5-ALA administration has been reported [29]. Blood concentrations of Pp IX peak approximately 4–6h after oral administration [30], coinciding with the timing of surgery, which may explain the pronounced vasodilation during this period. The vasodilatory effects of anesthetic drugs likely exacerbate this phenomenon. Our findings indicate that withdrawing antihypertensive medications on the morning of surgery does not effectively prevent intraoperative hypotension. Instead, clinical strategies such as optimizing the timing of antihypertensive withdrawal or prophylactic vasopressor administration should be explored.

The results of this study showed that the incidence of intraoperative hypotension was high in both groups. Therefore, it seems that clinical anesthesiologists should have vasopressors available in case of hypotension. In our cases, noradrenaline was needed to treat hypotension in some cases. On the other hand, the findings in this study do not allow us to conclude whether prophylactic administration of vasopressors could be better or not. Further research is needed to determine the efficacy of prophylactic vasopressors administration in patients administered 5-ALA.

The secondary outcomes did not show any significant differences in incidence between the two groups. However, it was noted that a small number of patients exhibited symptoms suggestive of hypotension or hypotension while in the ward. Regardless of whether antihypertensive medications were continued, there was a risk of hypotension during the period from administrating

5-ALA to entering the operating room. Blood pressure in the ward was monitored at infrequent intervals, which might result in undetected hypotension. Consequently, this could lead to adverse events such as syncope or dizziness, posing a potential risk to patient safety. To prevent such incidents, recognizing the risk of hypotension associated with 5-ALA ingestion by the ward staff is essential. To mitigate these risks, we recommend frequent blood pressure monitoring and considering preoperative fluid loading in patients administered 5-ALA.

The incidence of intraoperative hypotension after oral 5-ALA administration in this study was higher than the 0.8% reported in a phase III study [1,31]. This may be due to differences in patient characteristics. Prior studies have reported age and cardiovascular disease, including hypertension, as risk factors for hypotension after 5-ALA administration [9,10,13]. The patients in the two phase III studies were younger than those in this study (mean ages of 68.1 and 70.1 years old vs 75.8 years old). Moreover, the phase III study excluded patients with comorbidities, such as poorly controlled diabetes, hypertension, and arrhythmias. Differences in patient demographics and comorbidities likely contributed to the higher incidence of hypotension in this study. This emphasizes the importance of patient selection and stratification in future trials. Previous studies in clinical practice have defined hypotension variously, and the reported frequency of hypotension has varied from 6–70% [9,11,13]. The incidence of hypotension in this study does not deviate significantly from that in previous studies; therefore, we consider that the settings in this study are not far from common situations.

The fact that approximately half of the patients were classified as hypotensive was a point that we considered. As presented in Table 3, most anesthesiologists used ephedrine and phenylephrine to attempt controlling blood pressure and administered only a few catecholamines such as noradrenaline. Of the cases that presented with hypotension, 87.3% used some type of vasopressor during surgery; 85.2% of the non-hypotensive cases also did. The dose of ephedrine (12.0 (0.0–20.0) vs. 4.0 (0.0–10.0)) and phenylephrine (0.3 (0.0–0.94) vs. 0.1 (0.0–0.5)) tended to be higher in patients who presented with hypotension, which might indicate that ephedrine or phenylephrine use was not fully effective despite its repeated administration. The limited efficacy of ephedrine and phenylephrine suggests the need for alternative strategies, such as using noradrenaline as a first-line vasopressor in patients receiving 5-ALA.

This study has several strengths. First, it examined the association between antihypertensive use on the day of surgery and intraoperative hypotension while adjustment for confounding factors. For instance, patients with advanced age or cardiovascular disease may require individualized adjustments, as these factors significantly influence blood pressure dynamics. Our results showed that the crude results differed from the adjusted results, demonstrating the impact of confounding factors that might have affected the validity of direct comparisons without appropriate adjustment. Therefore, confounding adjustment should be performed to ensure accurate interpretation of the relationship. Second, the definition of hypotension used in this study was clinically meaningful and aligned with established criteria for organ perfusion and prognosis.

However, this study also has limitations. This study is limited by its retrospective design, imbalances in variables after weighting, and insufficient sample size for subgroup analyses. Furthermore, the lack of standardized protocols for blood pressure management may have introduced variability in outcomes. This is a common limitation of retrospective observational studies. Moreover, the generalizability of our findings is limited because most patients in our study underwent general anesthesia, whereas TURBT is sometimes performed under spinal anesthesia in other settings.

## Conclusions

This study found no evidence to support discontinuing antihypertensive medications on the morning of surgery and underscores the need for individualized perioperative strategies based on drug characteristics and patient profiles.

## Supporting information

**S1 Fig. Standardized mean difference of variables before and after propensity score weighting aimed to estimate the average treatment effect on overlap population. Red dots represent crude standardized mean difference and blue dots represent standardized mean difference after weighting.**
(TIF)

**S1 Table. Incidence of intraoperative hypotension of average treatment effect in the overlap population model. ATO, average treatment effect for the overlap population.**
(DOCX)

## Acknowledgments

We would like to thank Editage (www.editage.jp) for English language editing.

## Presentation

The preliminary data for this study was presented at the Japanese Society of Anesthesiologists Annual Meeting, which took place from June 6th to 8th, 2024 in Kobe, Japan.

## Author contributions

**Conceptualization:** Yuriko Kondo, Takahiro Mihara, Nanako Yoshikawa, Noriyuki Echigo, Yusuke Nagamine, Takahisa Goto.

**Data curation:** Yuriko Kondo, Takahiro Mihara.

**Formal analysis:** Yuriko Kondo, Takahiro Mihara.

**Investigation:** Yuriko Kondo, Takahiro Mihara, Nanako Yoshikawa.

**Methodology:** Yuriko Kondo, Takahiro Mihara, Nanako Yoshikawa, Noriyuki Echigo, Yusuke Nagamine, Takahisa Goto.

**Project administration:** Yuriko Kondo, Noriyuki Echigo.

**Supervision:** Takahiro Mihara.

**Validation:** Takahiro Mihara.

**Visualization:** Yuriko Kondo.

**Writing – original draft:** Yuriko Kondo.

**Writing – review & editing:** Takahiro Mihara, Nanako Yoshikawa, Noriyuki Echigo, Yusuke Nagamine, Takahisa Goto.

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
