## [Decision Letter · Decision Letter 0]

14 Aug 2024

PONE-D-24-24249Preoperative antihypertensives and hypotension during bladder tumor resection with oral 5-aminolevulinic acid administration.PLOS ONE

Dear Dr. Mihara,

Thank you for submitting your manuscript to PLOS ONE. After careful consideration, we feel that it has merit but does not fully meet PLOS ONE’s publication criteria as it currently stands. Therefore, we invite you to submit a revised version of the manuscript that addresses the points raised during the review process. Consider all the issues mentioned in the reviewers' comments when revising your manuscript. Outline every change made in response to their comments and provide suitable rebuttals for any comments not addressed. Please be aware that your revised submission may undergo re-review.

We look forward to receiving your revised manuscript.

Kind regards,

André Prato Schmidt, MD, PhD, MSc, MBA, DESAIC

Academic Editor

PLOS ONE

**Comments to the Author**

1. Is the manuscript technically sound, and do the data support the conclusions?

Reviewer #1: Yes

Reviewer #2: Yes

Reviewer #3: Yes

Reviewer #4: Yes

Reviewer #5: Yes

Reviewer #6: Yes

2. Has the statistical analysis been performed appropriately and rigorously? 

Reviewer #1: Yes

Reviewer #2: Yes

Reviewer #3: Yes

Reviewer #4: Yes

Reviewer #5: Yes

Reviewer #6: Yes

3. Have the authors made all data underlying the findings in their manuscript fully available?

Reviewer #1: Yes

Reviewer #2: Yes

Reviewer #3: Yes

Reviewer #4: Yes

Reviewer #5: Yes

Reviewer #6: Yes

4. Is the manuscript presented in an intelligible fashion and written in standard English?

Reviewer #1: Yes

Reviewer #2: Yes

Reviewer #3: Yes

Reviewer #4: No

Reviewer #5: Yes

Reviewer #6: Yes

5. Review Comments to the Author:

Reviewer #1: This is a fascinating study that examines the pros and cons of antihypertensive drug withdrawal in photodynamic diagnostic TURBT with oral 5-ALA. However, I think that the number of patients is too small to prove the superiority and non-inferiority of the intervention of discontinuing antihypertensives, as the authors state in their limitation. Also, as the authors state, the half-life of the antihypertensive drugs was not taken into account, and simply stopping them on the day of surgery may not have made a difference. From the above, the results of this study provide some insight, but unfortunately, I do not think we can make the judgment that "withdrawal of antihypertensive drugs on the day of surgery is meaningless". Further additional study on this topic is warranted.

Regarding patient background

Around 2019 (when the paper Nohara T et al. Int J Urol 26(11), 1064-1068, 2019 was published), intraoperative hypotension due to oral 5-ALA began to attract attention. After March 2020, there seems to be a sharp increase in cases of antihypertensive drug withdrawal on the day of surgery. Is this a departmental policy or an individual decision? We would like to know the reason for this.

Definition of hypotension

The primary outcome measure was the incidence of intraoperative hypotension. Hypotension was defined as MBP of less than 65 mmHg for ≥20 min, Although this definition was meaningful, the explanation of the definition was not well described. For example, if MBP ≥ 65 mmHg is reached even once during MBP< 65 mmHg for 20 min, due to bolus administration of vasoactive agents or fluid infusion, which would you consider there is a hypotensive event or not? Furthermore, what about intervals between blood pressure measurements for 20 minutes? The definition should be described in more detail.

Regarding RAS inhibitor

A previous study showed that regular ARB use was a risk for perioperative hypotension (defined as SBP < 80 mmHg at induction of anesthesia with hypotension) even if the ARB was withdrawn on the day of surgery (Nohara T et al. Int J Urol 26(11), 1064-1068, 2019). Is it possible that hypotensive events occur more frequently in patients who regularly use RAS inhibitors?

Regarding anesthesia.

Why do almost all patients at your institution receive general anesthesia? Is there any reason?

Conclusion

The author concluded that no substantial justification was provided for routinely discontinuing antihypertensive medications. I would like to know what the authors consider from this result. For example, “antihypertensives should be withdrawn on the day of surgery,” “it should be continued,” “whether to continue or discontinue antihypertensive medication should be considered in each individual case,” or “further analyses are warranted.“ please consider describing your opinion.

Reviewer #2: This is an intriguing study that evaluated the relationship between preoperative antihypertensive medication and hypotension during PDD-TURBT with oral 5-ALA administration. The assessment was conducted by adjusting for background factors using the overlap weighting method. The authors presented results indicating that, after adjusting for background factors, the presence or absence of preoperative antihypertensive medication on the day of surgery does not affect the incidence of intraoperative hypotension. This finding is considered to be highly beneficial in clinical practice.

Please address the following points:

1. The use of both the ATT and ATO models might be causing confusion. It would be better to use only one of these models.

2. Tables 1, 3, and 4: Please present the percentages for the Discontinued group and the Continued group in the adjusted model.

3. Discussion: The authors suggest that one reason for the lack of a significant association between the continuation or discontinuation of antihypertensive drugs on the morning of surgery and the incidence of intraoperative hypotension is the residual effects of the antihypertensive drugs, which may be correct. However, it is also possible that the hypotensive effect of 5-ALA is so strong that it overrides whether or not antihypertensive drugs are taken preoperatively. Previous studies have shown mixed results, with some indicating that the use of antihypertensive drugs is related to the development of hypotension and others indicating no such relationship. Ideally, this study might have been better if it had included a comparison among three groups, including a group of patients who were not taking antihypertensive medications. At the very least, the authors need to address this issue in the Discussion section.

Reviewer #3: This study examined the effect of antihypertensive medications on hypotension in patients taking ALA before surgery, defining hypotension as a mean blood pressure (MBP) of less than 65 mmHg for 20 minutes or longer. The authors conclude that the adjusted results suggest no significant association between the continuation of antihypertensive medications and the incidence of intraoperative hypotension to justify routine discontinuation of antihypertensive medications, but this alone does not seem to be novel. Therefore, please consider the following points.

Major points

• Although you report no significant association between the continuation of antihypertensive medication and the incidence of intraoperative hypotension, was there an association between the continuation of antihypertensive medication and the response to boosting medication after the occurrence of hypotension? Please add that point to the manuscript and add a new table if necessary.

Minor points.

• There have been several reports of ALA-induced hypotension from urology and anesthesiology departments, with varying definitions of hypotension. Why is hypotension defined as a mean blood pressure (MBP) of less than 65 mmHg for more than 20 minutes? Please include a brief explanation of why in your manuscript.

Reviewer #4: The authors analysed the frequency of hypotension in the continuation and discontinuation of antihypertensive medication before TURBT. This paper is considered valuable in demonstrating that continuation of antihypertensive medication is feasible. My opinion is as follows.

Major.

What is the time between antihypertensive drug administration and TURBT? Is there an association between that time and hypotension? Is it conceivable, for example, that hypotension is less likely to occur if antihypertensive drugs are taken only in the morning and surgery is performed in the afternoon?

Is the definition of hypotension appropriate? Can hypotension with various risks be left untreated for 20 minutes?

It is stated that the assessment of preoperative blood pressure in the 2ndary Outcome was not performed due to many missing values, but if it is a retrospective observational study, there is no need to include it in the methods section in the first place.

Minor.

What is the R ver?

Figure 1 - Not appropriate as there are two similar statements in the same block.

If the two groups are adjusted for patient background, consider presenting pre- and post-adjustment data.

There is no mention of hepatotoxicity, the most likely adverse effect of ALA.

Reviewer #5: Thank you very much for giving me the opportunity to review this manuscript. The study clarified the effect of antihypertensive drugs on ALA-related hypotension during surgery. The manuscript is very well written and scientifically solid, but I have raised some questions that should be addressed.

Major:

1. Line 96

The definition of hypotension (MBP less than 65 mmHg for > 20 min) is a quite severe situation. How did you manage the hypotension during surgery? According to the Table 3, about 60% of patients were administered with vasoactive agents including ephedrine. Do you mean MBP were less than 65 mmHg even in those patients? If so, the vasoactive agents seem not working for those patients.

As the management of hypotensive patients after PDD-ALA is important for the readers, please describe how the BP fluctuated during surgery.

Minor:

1. Line 215

91.4+17.6=109. Please correct the number.

Reviewer #6: This manuscript studied the effect of continuing antihypertensives on intraoperative hypotension in patients who underwent TURBT with 5-ALA administration. Hypotension is one of major concerns related to 5-ALA administration. In this study, the adjusted results suggested no significant association between the continuation of antihypertensive medication and the incidence of intraoperative hypotension. The manuscript was well written and revealed attractive results for urologists and anesthesiologists in the clinical practice.

I have several questions and comments to this manuscript as follows:

Major

1. This retrospective study included a total of 132 cases (108 patients) for the analysis. It seemed to be relatively small to make a conclusion. The authors may want to increase the number of patients or perform an external validation.

2. Based on the results of this study, the authors should show the clinical implication. Should we stop or continue the antihypertensives on the morning of surgery?

3. The authors concluded that “no substantial justification was provided for routinely discontinuing antihypertensive medications.” As the authors discussed in the manuscript, discontinuation on the morning of surgery may not be enough to eliminate the effects of antihypertensive drugs. Because the discontinuation period might affect the intraoperative hypotension, the authors should modify the conclusions as limited to the drug withdrawal on the morning of surgery.

Minor

1. Please include the number of regular used antihypertensives for each group in table 1.

2. Please indicate the type of continued antihypertensives for the continued group in table 1.

6. PLOS authors have the option to publish the peer review history of their article (what does this mean? ). If published, this will include your full peer review and any attached files.

**Do you want your identity to be public for this peer review?** For information about this choice, including consent withdrawal, please see our Privacy Policy .

Reviewer #1: **Yes: ** Takahiro Nohara

Reviewer #2: No

Reviewer #3: No

Reviewer #4: **Yes: ** Motohiro Fujiwara

Reviewer #5: No

Reviewer #6: No

---

## [Author Response · Author response to Decision Letter 1]

4 Nov 2024

Please refer the attached "Response to Reviewers" file.

---

## [Editor Report · Decision Letter 1]

22 Nov 2024

PONE-D-24-24249R1Preoperative antihypertensives and hypotension during bladder tumor resection with oral 5-aminolevulinic acid administration.PLOS ONE

Dear Dr. Mihara,

Thank you for submitting your manuscript to PLOS ONE. After careful consideration, we feel that it has merit but does not fully meet PLOS ONE’s publication criteria as it currently stands. Therefore, we invite you to submit a revised version of the manuscript that addresses the points raised during the review process.

We look forward to receiving your revised manuscript.

Kind regards,

André Prato Schmidt, MD, PhD, MSc, MBA, DESAIC

Academic Editor

PLOS ONE

Journal Requirements:

Additional Editor Comments:

This is a revised version of a manuscript describing a single-center observational study examines the association between the continuation of antihypertensive medications on the morning of surgery and the incidence of intraoperative hypotension in patients undergoing transurethral resection of bladder tumors (TURBT) after receiving 5-aminolevulinic acid hydrochloride (5-ALA). The study included 132 cases, divided into two groups: those who continued taking antihypertensives and those who discontinued them. Propensity score weighting was used to adjust for confounding factors, and the primary outcome was defined as hypotension with a mean blood pressure (MBP) below 65 mmHg for at least 20 minutes. The study found no significant association between the continuation of antihypertensive medications and intraoperative hypotension after adjustment. The findings suggest that routine discontinuation of antihypertensives may not be necessary but emphasize the need for further research to consider drug type, half-life, and discontinuation periods.

Again, the study addresses a clinically relevant question, particularly in the context of preoperative management for patients undergoing TURBT after 5-ALA administration. The use of propensity score weighting improves the robustness of the analysis by adjusting for potential confounding factors. The inclusion of a clinically meaningful definition of hypotension provides practical value to anesthesiologists and perioperative clinicians.

The study has overall some known limitations, the observational design limits causal inferences and is subject to selection bias, imbalances in variables like eGFR and stroke rates after propensity score adjustment could impact the interpretation of results and the lack of standardized intraoperative blood pressure management protocols and vasopressor administration adds variability to the outcomes.

The authors addressed most of the reviewers' comments effectively and improved the manuscript in its revised form. However, some areas, such as the discussion on intraoperative management variability, drug-specific effects, and selection bias, could benefit from further elaboration. Additionally, emphasizing the clinical significance of the adjusted versus crude results and expanding on secondary outcomes would be interesting to readers. Please find below a few suggestions for the next version of this manuscript.

1. The manuscript reports the crude incidence of hypotension and adjusted results but does not elaborate on the clinical implications of these differences. A clearer discussion on why the crude incidence differs significantly while the adjusted incidence does not would strengthen the interpretation.

2. The secondary outcomes, such as postoperative adverse events, are briefly mentioned but not explored in detail. Briefly expanding this discussion could provide additional insights into the clinical implications of the findings.

3. Clarify the timeline of antihypertensive discontinuation in more detail, particularly for long-acting drugs, to better contextualize the results.

4. Please highlight practical recommendations, such as the need for prophylactic vasopressors, as actionable conclusions for clinicians.

5. The discussion could be slightly improved in a language context. Please find below some suggestions in order to improve clarity in parts of your Discussion section:

“In this study, no significant association was found between the continuation or discontinuation of antihypertensive medications on the morning of surgery and the incidence of intraoperative hypotension in adult patients undergoing scheduled TURBT after 5-ALA administration. Additionally, the incidence of adverse events during hospitalization did not differ significantly between the two groups. These findings suggest that there is no substantial justification for routinely discontinuing antihypertensive medications on the morning of surgery.”

“One possible explanation for this result is the residual effect of long-acting antihypertensive drugs. Regular intake of antihypertensives with long half-lives, even when discontinued on the morning of surgery, may result in persistent pharmacologic effects during the perioperative period. Previous studies have identified long-acting ACE inhibitors (ACE-Is) and angiotensin receptor blockers (ARBs) as risk factors for hypotension, even after a 24-hour withdrawal period [26,27]. For example, severe hypotension has been reported in patients taking amlodipine, a calcium channel blocker with a half-life of approximately 40 hours, despite discontinuation on the morning of surgery [27]. In our study, more than half of the patients were on amlodipine or combination drugs containing amlodipine, while a smaller proportion were taking telmisartan, an ARB with a long half-life. For patients regularly using long-acting antihypertensives, a withdrawal period of 12 to 24 hours may be insufficient to eliminate their effects on intraoperative blood pressure. Future research should evaluate the impact of drug type, half-life, and timing of withdrawal on the risk of hypotension.”

“Another potential factor is the potent hypotensive effect of 5-ALA itself, which may override the influence of preoperative antihypertensive medication. While our study was not designed to evaluate this interaction specifically, future research should explore this mechanism further. The pathogenesis of hypotension following 5-ALA administration remains unclear, but animal studies suggest that its metabolite, protoporphyrin IX (Pp IX), activates soluble guanylate cyclase, leading to vasodilation of systemic and pulmonary vessels. Additionally, increased vascular permeability after 5-ALA administration has been reported. Blood concentrations of Pp IX peak approximately 4–6 hours after oral administration, coinciding with the timing of surgery, which may explain the pronounced vasodilation during this period. The vasodilatory effects of anesthetic drugs likely exacerbate this phenomenon. Our findings indicate that withdrawing antihypertensive medications on the morning of surgery does not effectively prevent intraoperative hypotension. Instead, clinical strategies such as optimizing the timing of antihypertensive withdrawal or prophylactic vasopressor administration should be explored.”

“This study has several strengths. First, it examined the association between antihypertensive use on the day of surgery and intraoperative hypotension while adjusting for confounding factors. Second, the definition of hypotension used in this study was clinically meaningful and aligned with established criteria for organ perfusion and prognosis.”

“However, the study also has limitations. Imbalances in variables such as eGFR and stroke rates persisted after weighting, although these imbalances are unlikely to have overturned the results. The association between antihypertensive use and hypotension may differ depending on the type and half-life of the medications, but the study's sample size was insufficient for subgroup analyses. Additionally, the lack of information on the timing of the last antihypertensive dose precluded a detailed investigation of drug pharmacokinetics. Selection bias may have occurred, as the reasons for continuing or discontinuing antihypertensives were not documented in all cases. Finally, the absence of a standardized protocol for blood pressure management and vasopressor use during surgery introduces variability in the outcomes. Moreover, the generalizability of our findings is limited because most patients underwent general anesthesia, whereas TURBT is often performed under spinal anesthesia in other settings.”

“Conclusion

This study found no significant association between the use of antihypertensive medications on the morning of surgery and the incidence of intraoperative hypotension. Consequently, there is no substantial justification for the routine discontinuation of antihypertensive medications on the day of surgery. Further research should consider drug type, half-life, and withdrawal timing to refine preoperative management strategies.”

---

## [Author Response · Author response to Decision Letter 2]

23 Dec 2024

Please find the attached "Response to Reviewers" file.

---

## [Editor Report · Decision Letter 2]

15 Jan 2025

PONE-D-24-24249R2Preoperative antihypertensives and hypotension during bladder tumor resection with oral 5-aminolevulinic acid administrationPLOS ONE

Dear Dr. Mihara,

Thank you for submitting your manuscript to PLOS ONE. After careful consideration, we feel that it has merit but does not fully meet PLOS ONE’s publication criteria as it currently stands. Therefore, we invite you to submit a revised version of the manuscript that addresses the points raised during the review process.

We look forward to receiving your revised manuscript.

Kind regards,

André Prato Schmidt, MD, PhD, MSc, MBA, DESAIC

Academic Editor

PLOS ONE

Journal Requirements:

Additional Editor Comments:

The manuscript presents a well-structured and relevant study addressing the effects of preoperative antihypertensive medication use on intraoperative hypotension in patients undergoing TURBT after 5-ALA administration. It makes significant contributions to understanding perioperative management strategies in these cases. The use of propensity score weighting strengthens the analysis by addressing confounding factors. However, minor improvements can be made to enhance clarity and reduce redundancy.

1. Table 1 content overload: Table 1 contains an overwhelming amount of demographic and clinical information. While comprehensive, it hinders readability and distracts from the key findings. Please restrict Table 1 to the most relevant demographic and clinical variables (e.g., age, sex, BMI, ASA-PS, type of antihypertensives, etc). Move less critical variables to a supplementary table.

2. Table 3 should include only vasoactive drugs that were actually used in the studied population. Please correct and provide adequate legends for this table.

3. The discussion is comprehensive but can be better organized. Consider grouping similar ideas together for improved readability. Suggestions for specific corrections are provided below:

- Line 265-266: "No significant association was found between continuing or discontinuing antihypertensive medications on the morning of surgery and intraoperative hypotension during scheduled TURBT following 5-ALA administration."

- The phrase "persistent pharmacologic effects during the perioperative period" is unclear without elaboration. Specify the clinical relevance of these persistent effects. For example: "...result in persistent pharmacologic effects during the perioperative period, such as attenuation of blood pressure fluctuations."

- Line 286-289: The description of withdrawal periods is informative but repetitive. Condense the explanation: "Withdrawal periods varied between patients, with some experiencing approximately 28 hours and others about 12 hours, depending on their usual medication schedule."

- Line 296-302: Lack of clarity regarding the hypothesis of 5-ALA effects. Explicitly state how this impacts clinical decisions: "These findings suggest that the vasodilatory effects of 5-ALA, compounded by anesthetic drugs, may outweigh the influence of preoperative antihypertensive withdrawal. This highlights the need for tailored strategies, such as timing adjustments or pharmacologic interventions."

- Line 321-329: Blood pressure monitoring is discussed but lacks actionable clinical recommendations. Include specific strategies: "To mitigate these risks, we recommend frequent blood pressure monitoring and considering preoperative fluid loading in patients administered 5-ALA."

- Line 331-339: The comparison to the phase III study could emphasize its implications. Provide a clearer explanation: "Differences in patient demographics and comorbidities likely contributed to the higher incidence of hypotension in this study. This emphasizes the importance of patient selection and stratification in future trials."

- Line 343-350: The efficacy of vasopressors is discussed but not thoroughly analyzed. Elaborate on why ephedrine and phenylephrine might be less effective and propose alternatives: "The limited efficacy of ephedrine and phenylephrine suggests the need for alternative strategies, such as using noradrenaline as a first-line vasopressor in patients receiving 5-ALA."

- Line 354-359: The statement about confounding factors is strong but lacks context. Include a specific example of a confounding factor: "For instance, patients with advanced age or cardiovascular disease may require individualized adjustments, as these factors significantly influence blood pressure dynamics."

- Line 363-371: The limitations section is detailed but can be more concise. Combine related points: "This study is limited by its retrospective design, imbalances in variables after weighting, and insufficient sample size for subgroup analyses. Furthermore, the lack of standardized protocols for blood pressure management may have introduced variability in outcomes."

- Line 377-382: The conclusion repeats earlier points. Focus on actionable insights: "This study supports the continued use of antihypertensive medications on the morning of surgery and highlights the need for individualized perioperative strategies based on drug characteristics and patient profiles."

---

## [Author Response · Author response to Decision Letter 3]

27 Jan 2025

Please find the attached "Response to Reviewers" file.

---

## [Editor Report · Decision Letter 3]

3 Feb 2025

Preoperative antihypertensives and hypotension during bladder tumor resection with oral 5-aminolevulinic acid administration

PONE-D-24-24249R3

Dear Dr. Mihara,

We’re pleased to inform you that your manuscript has been judged scientifically suitable for publication and will be formally accepted for publication once it meets all outstanding technical requirements.

Kind regards,

André Prato Schmidt, MD, PhD, MSc, MBA, DESAIC

Academic Editor

PLOS ONE

---

## [Editor Report · Acceptance letter]

PONE-D-24-24249R3

PLOS ONE

Dear Dr. Mihara,

I'm pleased to inform you that your manuscript has been deemed suitable for publication in PLOS ONE. Congratulations! Your manuscript is now being handed over to our production team.

Kind regards,

on behalf of

Dr. André Prato Schmidt

Academic Editor

PLOS ONE